# Chemical Composition, Antimicrobial and Antioxidant Bioautography Activity of Essential Oil from Leaves of Amazon Plant *Clinopodium brownei* (Sw.)

**DOI:** 10.3390/molecules28041741

**Published:** 2023-02-11

**Authors:** Paco Noriega, Lissette Calderón, Andrea Ojeda, Erika Paredes

**Affiliations:** 1Master Program of Natural Pharmaceutical Products, Universidad Politécnica Salesiana, Avenida 12 de Octubre N2422 y Wilson, Quito 170109, Ecuador; 2Group of Research and Development in Science Applied to Biological Resourses, Universidad Politécnica Salesiana, Avenida 12 de Octubre N2422 y Wilson, Quito 170109, Ecuador

**Keywords:** *Clinopodium brownei*, caryophyllene, cinnamic acid derivates, bioactivity bioautography

## Abstract

The Amazonian region of Ecuador has an extremely rich vegetal biodiversity, and its inhabitants have proven to have a millennial ancestral knowledge of the therapeutic and medicinal use of these resources. This work aimed to evaluate the chemical composition and biological activity of the essential oil obtained from the medicinal plant *Clinopodium brownei* (Sw.) Kuntze, which is widely spread in tropical and subtropical America. This species is traditionally used for treating respiratory and digestive diseases and is also known for its analgesic properties. Most of the molecules detected on a non-polar column were ethyl cinnamate 21.4%, pulegone 20.76%, methyl cinnamate 16.68%, caryophyllene 8.17%, β-selinene 7.92% and menthone 7.51%, while those detected on a polar column were: pulegone 29.90%, ethyl cinnamate 18.75%, methyl cinnamate 13.82%, caryophyllene 10.0% and menthone 8.04%. The antioxidant activity by the assays, DPPH (2.2-diphenyl-1-picrylhydrazyl) and ABTS (2.2′-azino-bis (3-ethylbenzothiazoline-6-sulfonic acid)), shows the following values of 50% inhibition of oxidation, IC50 DPPH 1.77 mg/mL, IC50 ABTS 0.06 mg/mL, which, compared to the essential oil of *Thymus vulgaris* (natural positive control), turn out to be less active. Bioautography indicates that the molecules responsible for the antioxidant activity are derived from cinnamic acid: ethyl cinnamate and methyl cinnamate, and caryophyllene. The antimicrobial activity on the nine microorganisms evaluated shows bacterial growth inhibitory concentrations ranging from 13.6 mg/mL for *Staphylococcus epidermidis* ATCC 14990 to 3.1 mg/mL for *Candida albicans* ATCC 10231; the results are lower than those of the positive control. Bioautography assigns antimicrobial activity to caryophyllene. The results indicate a very interesting activity of the essential oil and several of its molecules, validating the traditional use and the importance of this medicinal plant from Ecuador.

## 1. Introduction

The tropical and subtropical regions have the most significant biological biodiversity on the planet; a large part of the Ecuadorian territory is in tropical environments, except for the Andean mountains, which have a climate that ranges from temperate to cold in the highest parts [1,2]. Due to this peculiar biogeography of Ecuador, its biodiversity is considered one of the most important in the world, and it is included in the list of 17 megadiverse countries. In Ecuador, there are approximately 25.000 plant species, representing about 10% of all plants in the world, despite occupying only 0.06% of the global land surface [3].

There are many medicinal plants in the tropics that are used for treating various diseases and symptoms [4,5,6,7,8,9,10,11], representing a source of secondary metabolites. Essential oils are among the secondary metabolites with the most significant potential and industrial interest due to their simple extraction method [12], the easy analysis method of their components [13], and their wide range of biological, pharmaceutical [14,15,16], and cosmetic applications [17,18].

Essential oils of several species from the tropical zones of Ecuador have been investigated, obtaining significant results, highlighting the evaluations of chemical composition and bioactivity; among the most interesting plants are *Ocotea quixos* [19,20,21], *Piper aduncum* [22,23], *Piper carpunya* [24,25], *Siparuna aspera* and *Siparuna macrotepala* [26], among others.

Among the Ecuadorian clinopidum species, *Clinopidium nubigenum* stands out for its antimicrobial respiratory activity [27]; and *Clinopodium tomentosum* is anti-inflammatory [28]. *Clinopodium brownei*, an aromatic, herbaceous species, is found in the southeastern United States, Central America, the Caribbean, and South America; in Ecuador, it is found in tropical and subtropical areas of several provinces [27]. It is traditionally known as “tipo pequeño”, “warmi poleo” and “poleo de la tierra”; it is used for treating respiratory diseases, diseases of the digestive system, dental cavities, and fever; it is also known for its analgesic properties [28,29,30]. Chemical studies of its essential oil show the presence of pulegone and menthone as the most abundant molecules, with smaller proportions of isomenthone, isopulegone, β-pinene, caryophyllene, and calcorene [27,31,32]. In-vitro assays of antioxidant and antimicrobial activity have been conducted with good results on its essential oil [26,27,28,29,30,31,32]. All the information on its ancestral uses and preliminary studies on *C. brownei* identify the plant as an excellent medicinal resource. For this reason, this research aims to evaluate the chemical compounds and bioactivity in the essential oil for pharmaceutical and cosmetic use.

## 2. Results

### 2.1. Clinopodiun Brownei Essential Oil Extraction

The average yield of the essential oil was 0.25 ± 0.01% *w*/*w*. Different extraction assays of *C. brownei* essential oil show yields ranging from 0.58% to 0.12% [29,33,34].

### 2.2. Chemical Composition

Twenty-four compounds were detected with the Termo Scientific TR-5MS (5%-phenyl-95% dimethyl) polysiloxane) column, of which 18 were identified for a total of 98.52%. The most abundant molecules were ethyl cinnamate 21.4%, pulegone 20.76%, methyl cinnamate 16.68%, caryophyllene 8.17%, β-selinene 7.92%, and menthone 7.51%. Using the Agilent DBWax (polyethylene glycol) column, 22 molecules were detected, and 17 were identified, for a total of 98.35%. The most abundant components were pulegone 29.90%, ethyl cinnamate 18.75%, methyl cinnamate 13.82%, caryophyllene 10.0%, and menthone 8.04%. Table 1 shows the chemical composition of the oil with the two columns used. Appendix A show the chromatograms for each analysis.

### 2.3. Antioxidant Activity

The results of antioxidant activity using the DPPH and ABTS methods, evaluated as the inhibition percentage at 50%, can be seen in Table 2. The IC50 values were calculated from the variation in inhibition percentage for each concentration of essential oil in contact with the radicals DPPH and ABTS. The essential oil of *T. vulgaris* (natural control) [19] and the chemical control butylated hydroxyanisole (BHA) were used as activity standards, showing the differences in activity with the natural positive standard and the chemical control.

### 2.4. Bioautography Antioxidant

Chromatographic separation on HP-TLC plates and subsequent disclosure with DPPH reagent showed three components with antioxidant activity at the following retention factors: (Rf), 0.633, 0.709, and 0.937. Verification of their chemical identity by mass spectrometry yielded the following results: Rf 0.633 (methyl cinnamate), Rf 0.709 (ethyl cinnamate), and Rf 0.937 (caryophyllene). Figure 1 shows the individual activity with their antioxidant molecule. The mass spectrums of activity molecules can be appreciated in the Appendix A.

### 2.5. Antimicrobial Activity

The minimum inhibitory concentration results (MIC), evaluated in 4 Gram+ strains, 4 Gram-strains, and one yeast, are shown in Table 3 and Figure 2. The values are compared with the MIC values of *T. vulgaris* oil, known for its high antimicrobial activity.

### 2.6. Bioautography Antimicrobial

Bioautographic assays confirm the antimicrobial activity in a single molecule identified in the retention factor (Rf = 0.937), which corresponds to E-caryophyllene. Since the separation occurred in decreasing volumes of *C. brownei* essential oil solutions (25, 20, 15, 10, and 5 microliters), a higher activity on Gram-positive bacteria is observed. These results are shown in Figure 3.

## 3. Discussion

There are similarities and differences in the evaluation of the chemical composition of the essential oil. In all investigations, including this one, the percentage for pulogene and menthona is significant [29,33,34]. However, in our research, a high content of two esters derived from cinnamic acid, methyl cinnamate and ethyl cinnamate, can be observed. This suggests that the oil of this research can be considered a chemotype with a high percentage of cinnamic acid derivatives.

The antioxidant activity reported was lower than that observed in *T. vulgaris* oil (positive control). The only reference research on *C. brownei* oil, carried out in Colombia [34], proposes IC_50_ values between a range of 2.5 to 10 mg/mL, much less active than those found in our research, whose values are IC_50_ DPPH is 1.771 mg/mL, IC_50_ ABTS of 0.060 mg/mL. When the IC_50_ values are lower, the antioxidant activity is higher. This increase can be explained by the presence of cinnamic acid derivatives, which in the bioautographic assays, together with caryophyllene, proved to be the active antioxidant molecules. The high antioxidant activity of methyl cinnamate [37] and ethyl cinnamate [38] has been demonstrated in individual studies. Several studies on caryophyllene have evaluated its antioxidant, anti-inflammatory, and anticarcinogenic activity [39,40,41].

The activity in nine microorganisms shows minimum inhibitory concentration values ranging from 13.57 mg/mL for S. epidermidis to 3.11 mg/mL for C. albicans. The activity of the natural positive control is higher in all the tests; however, if values described in the scientific literature for known essential oils, such as chamomile [42], cinnamon [43], and Rosmarinus [44], are analyzed, the results are quite interesting. Comparing this research with another carried out in Venezuela, the results are quite similar for common bacteria: *Staphylococcus aureus* (7.921 mg/mL research in Ecuador and 6.250 mg/mL research in Venezuela); *Enterococcus faecalis* (5.540 mg/mL research in Ecuador and 3.125 mg/mL research in Venezuela); *Escherichia coli* (6.250 mg/mL research in Ecuador and 6.250 mg/mL research in Venezuela), and *Pseudomonas aeruginosa* (8.38 mg/mL research in Ecuador and 3.125 mg/mL research in Venezuela) [45]. Bioautography evidences the antimicrobial activity of caryophyllene; there is ample evidence of its action on this molecule, such as that performed on *P. gingivalis* [46]. Another research determines an inhibition value of 2.5% for *B. cereus* [47]. On the other hand, some essential oils in which caryophyllene is found in high concentrations have proven to have significant antimicrobial activity [48,49,50].

## 4. Materials and Methods

### 4.1. Plant Material

The plant material was collected in the parish of Rio Blanco, Morona Santiago, Ecuador, in January 2021, at the coordinates 2°28′54.0″ S and 78°09′03″ W. The plant was identified by botanist Marco Cerna, from the herbarium of Universidad Politécnica Salesiana.

### 4.2. Essential Oil Extraction

The extraction of the essential oil was carried out by hydrodistillation, using Clevenger equipment with 500 mL capacity. A hundred grams of fresh leaves, which were distilled for 3 h, were used in each distillation. The process was carried out in triplicate. Subsequently, the essential oil was stored in an amber bottle at a temperature of 4 °C.

### 4.3. GC/FID Analysis

Each individual molecule was quantified using GC/FID, triplicating each chromatographic test and calculating its relative area. A gas chromatograph Varian 3900, with FID detector, was used. The sample was prepared by diluting 10 µL of essential oil in 990 µL of dichloromethane. The operating conditions were as follows: analysis started at 45 °C, reaching 100 °C at a rate of 1 °C per minute; then, it reached 250 °C at a speed of 5 °C per minute, staying at this temperature for 15 min, for a total analysis time of 90 min. The haul gas was 99.9999% pure helium at a flow of 1 mL/min and split-ratio of 1:25. A chromatographic column Zebron ZB-5MS (5%-phenyl-95% dimethyl) polysiloxane) with a length of 30 m, a thickness of 0.25 mm and a film thickness of 0.25 m was used.

### 4.4. GC/MS Analysis

The first analysis was carried out with a Termo Scientific TR-5MS column (5%-phenyl-95% dimethyl) polysiloxane) with a length of 30 m, a thickness of 0.25 mm, and a film thickness of 0.25 m. A chromatograph Trace 1310 coupled to a mass spectrometer was used.

The operating conditions were as follows: analysis started at 60 °C for 5 min, reaching 100 °C at a rate of 2 °C per minute; then, it reached 200 °C at a speed of 3 °C per minute, after it reached 230 °C at a rate 5 °C per minute, staying at this temperature for 5 min, for a total analysis time of 60 min.

Mass spectrometer conditions were as follows: ionization energy: 70 eV; emission current: 10 µAmp; scan rate: 1 scan/s; mass range: 40–350 Da; trap temperature: 230 °C; transfer line temperature: 200 °C [26].

The second analysis was carried out with an Agilent DBWax column (polyethylene glycol) with a length of 20 m, a thickness of 0.10 mm, and a film thickness of 0.20 m. A chromatograph EVOQ 436 GC–TQ coupled to a mass spectrometer was used.

The operating conditions were as follows: analysis started at 50 °C for 2 min, reaching 80 °C at a rate of 5 °C per minute; then, it reached 250 °C at a speed of 5 °C per minute, staying at this temperature for 1 min, for a total analysis time of 49.50 min.

Mass spectrometer conditions were as follows: ionization energy: 70 eV; emission current: 10 µAmp; scan rate: 1 scan/s; mass range: 35–400 Da; trap temperature: 220 °C; transfer line temperature: 260 °C [51].

### 4.5. Identification of Compounds

The NIST 2001 Mass Spectra Database was used to identify molecules [52]. In addition, the retention index rates (RI) of each compound were calculated by comparing a series of C_8_–C_30_ alkanes, and comparing the theoretical arithmetic retention rates, contrasting them with the scientific literature of the Adams database for 5%-phenyl-95% dimethyl) polysiloxane) column [35], and the research of Babushok for Agilent DBWax column (polyethylene glycol) [36].

### 4.6. Antioxidant Activity, DPPH Assay

DPPH is a molecule that is already radicalized and directly allows measurement of the oxidation inhibition percentage [53]. The DPPH solution was prepared by weighing 40 mg and dissolving it in 100 mL of 96% ethanol. The essential oil of *C. brownei* and the natural control oil *T. vulgaris* were prepared by dissolving 100 µL of oil in 900 µL of absolute ethanol (99.9% purity). Butylated hydroxyanisole (BHA) was used as a positive chemical control at a concentration of 1 mg/mL. The reading was performed in a UV/VIS spectrophotometric reader for BIOTEK microplates, EPOCH model.

Using 96-well plates, the following volumes of the essential oil and BHA solutions were added to each well in increasing order: 0, 1, 2, 5, 10, 15, and 20 µL. It was added to each essential oil volume, DPPH solution to until reach 200 µL; the plates were shaken in the dark for 30 min, and their absorbance was read at a wavelength of 517 nm. The assay was performed in triplicate for each solution. The oxidation inhibition percentage was calculated according to the following equation:(1)$$\%\;inhibition\;DPPH=\left(\frac{blank\;absorbance-Sample\;absorbance}{blank\;absorbance}\right)\;\times \;100$$

With the data obtained from the calibration curve, the DPPH IC_50_ is calculated and compared with the controls used.

### 4.7. Antioxidant Activity, ABTS Assay

ABTS is a molecule that requires its radicalized form to be generated [54], which was carried out by mixing a solution of ABTS containing 1.1 mg/mL with 0.25 µL of a solution of K_2_S_2_O_8_ with 18.8 mg/mL concentration. The mixture was kept in the dark for 12 h, and its absorbance was adjusted to 0.700 ± 0.02. As in the DPPH assay, the essential oils of *C. brownei* and *T. vulgaris* (natural control) and BHA as a chemical control at a concentration of 1 mg/mL, were used in the assay.

The arrangement and preparation of the samples in the 96-well plates were the same as in the DPPH assay. The reading was taken 1 min after applying the radical ABTS and was determined at a wavelength of 734 nm in a UV/VIS spectrophotometer for BIOTEK microplates, EPOCH model.

The same formula and methodology, as described in the DPPH assay, Section 4.6, were used to calculate the inhibition percentages and the IC_50_ ABTS.

### 4.8. Antioxidant Bioautography

Several studies show that it is possible to identify the molecules responsible for biological activity in an essential oil using bioautographic techniques [21,26,55]. To separate the active compounds, a Merck silica gel HPTLC chromatographic plate was used, to which 25 µL of *C. brownei* oil dissolved in methanol at a concentration of 30 mg/mL was added. The mobile phase was composed of toluene, ethyl acetate and petroleum ether, in proportions of 97:7:20. Once the chromatographic plate was prepared, it was developed with DPPH at a concentration of 1% in ethanol, and the compounds that changed their coloration from violet to yellow were observed [21]. From these active fractions, the compounds were isolated by dissolving them in dichloromethane. The GC/MS assay was performed under the same conditions, as described in Section 4.4 for TR-5MS column.

### 4.9. Antimicrobial Activity

Assays were performed on 4 Gram-positive bacteria: *Staphylococcus aureus* ATCC 6328, *Enterococcus faecalis* ATCC 29212, *Listeria grayi* ATCC 1912, and *Staphylococcus epidermidis* ATCC 14990; 4 Gram-negative bacteria: *Escherichia coli* ATCC 25922, *Proteus vulgaris* ATCC 6380, *Klebsiella oxytoca* ATCC 8724 and *Pseudomonas aeruginosa* ATCC 9027; and one yeast *Candida albicans* ATCC 10231. The technique used to calculate the minimum inhibition concentration is known as broth micro dilution, which has been used in several studies of antimicrobial activity with essential oils [55,56].

Bacteria and yeast were reactivated in appropriate media according to growth specifications, reaching concentrations of 1 × 10^8^ CFU/mL for bacteria and 1 × 10^6^ CFU/mL for yeast. In a 96-well plate, 10 µL of *C. brownei* essential oil solution in DMSO (99.6%) + 150 µL of the medium with the microorganism, and 40 µL of the 2.5% TTC (Triphenyl tetrazolium chloride) dye were seeded. *T. vulgaris* essential oil was used as a positive control and DMSO as a negative control.

In the cases of the tested essential oils and the positive control, the concentration used is 5%, diluting it in descending ratios: 1:2, 1:4, 1:18, 1:16, 1:32, 1:64, and 1:128. Finally, plates were incubated for 24 h in agitation at a temperature of 35 ± 2 degrees Celsius for bacteria and for 48 h in agitation at a temperature of 25 ± 2 degrees Celsius for yeast. After this time, absorbance was measured at 615 nm in a UV/VIS spectrophotometer for BIOTEK microplates, EPOCH model.

### 4.10. Antimicrobial Bioautography

In this assay, two bacteria that needed the lowest concentrations of *C. brownei* essential oil to inhibit bacterial growth were selected; in this case: *Listeria grayi* ATCC 1912 (Gram-positive bacteria) and *Proteus vulgaris* ATCC 6380 (Gram-negative bacteria). The components of the essential oil were separated on a Merk silica gel HPTLC chromatographic plate; essential oil at a concentration of 30 mg/mL was incubated using a mobile phase composed of toluene, ethyl acetate and petroleum ether in proportions of 97:7:20. Disclosure was performed by covering the plate with culture medium, containing the target bacteria and the TTC dye. The plates were incubated for 24 h at 37 degrees Celsius, showing activity in those fractions that stain white [21]. Subsequently, the active fractions were extracted with dichloromethane and analyzed in a GC/MS, under the same conditions described in Section 4.4 for TR-5MS column.

### 4.11. Statistics

For all data collected, relative standard deviations and statistical significance (student’s test; *p* < 0.05), one-way ANOVA, and LSD post hoc Fisher’s honest significant difference test were taken where appropriate. All computations were collected using the statistical software STATISTICA 6.0.

## 5. Conclusions

The chemistry of the essential oil of *C. brownei* allows us to observe similarities and differences with the few studies previously carried out on the species. The most relevant result is that the Ecuadorian Amazonian variety contains high percentages of compounds derived from cinnamic acid, methyl cinnamate and ethyl cinnamate, both representing about 40% of all the oil, something that had not been observed in previous studies. Additionally, these molecules would be related to antioxidant activity as confirmed by bioautographic evaluations.

Another molecule involved in biological activity is caryophyllene, which affects both antimicrobial and antioxidant potential. This molecule has aroused pharmaceutical interest in recent times because it is a potential antimicrobial, antioxidant, anti-inflammatory, and anticarcinogenic molecule.

Based on the interesting chemical composition and biological activity, it is advisable to continue testing complementary biological activity, such as anti-inflammatory, cytotoxic, insect repellent, etc., thus having a validated biological resource that could be used as an alternative to treat various diseases and symptoms.

## Figures and Tables

**Figure 1 molecules-28-01741-f001:**
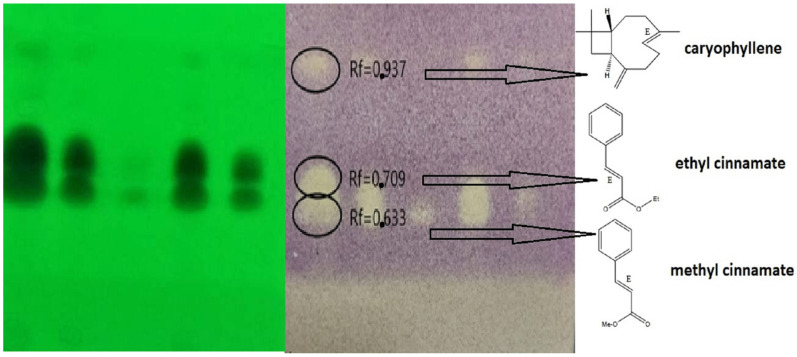
DPPH Bioautographic assay of *Clinopodium brownei* essential oil, bioactive molecules: Rf 0.633 methyl cinnamate, Rf 0.709 ethyl cinnamate and Rf 0.937 caryophyllene.

**Figure 2 molecules-28-01741-f002:**
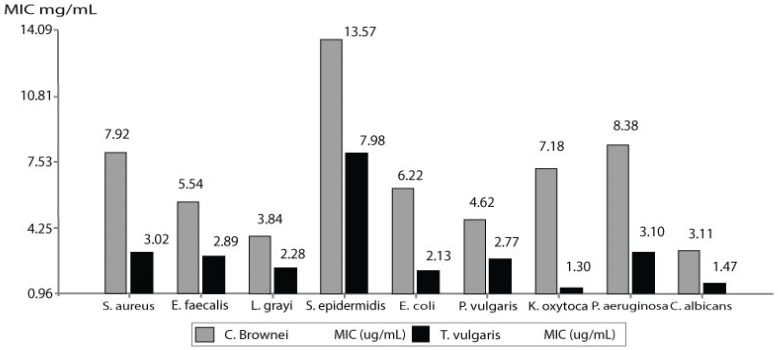
Graphical comparison of minimum inhibitory concentration (MIC) values between the essential oils of *C. brownei* and *T. vulgaris* (natural control). Mean ± SD (*n* = 3), *p* < 0.05.

**Figure 3 molecules-28-01741-f003:**
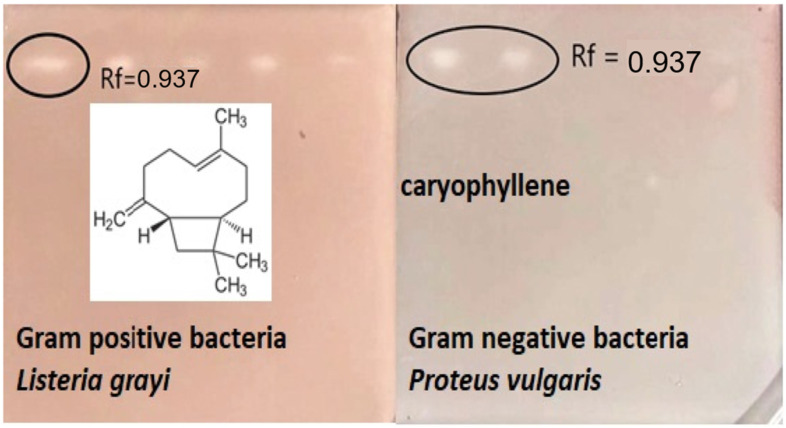
Bioautographic antibacterial assay of *Clinopodium brownei* essential oil, bioactive molecule Rf 0.937 caryophyllene.

**Table 1 molecules-28-01741-t001:** Chemical composition of the essential oil of *Clinopodium brownie*, using two chromatographic systems, with the polar and non-polar columns.

Components (Non-Polar Column)	TR5MS RI cal ^a^	RI Lit ^b^	%RDA	Components (Polar Column)	DB5-Wax RI cal ^c^	RT Lit ^d^	% RDA
3-octanone	992	979	0.24	3-octanone	1263	1255	0.22
menthone	1164	1148	7.51	2-pentanone, 4-hydroxy-4-methyl	1371	-	1.42
iso-isopulegol	1172	1159	1.18	menthone	1471	1465	8.04
neoiso.pulegol	1176	1167	0.49	α-copaene	1496	1491	3.07
pulegone	1252	1233	20.76	β-elemene	1594	1591	0.64
α-copaene	1376	1374	2.87	(E)-caryophyllene	1602	1599	10.0
(E)-methyl cinnamate	1401	1376	16.68	pulegone	1653	1655	29.9
(E)-caryophyllene	1421	1417	8.17	β-humulene	1672	1667	1.96
(E)-ethyl cinnamate	1465	1443	21.4	γ-gurjunene	1692	1668	0.49
β-humulene	1458	1436	2.17	α-terpinyl acetate	1697	1695	0.31
γ-gurjunene	1487	1475	0.81	β-selinene	1722	1717	6.46
β-selinene	1493	1489	7.92	α-selinene	1726	1725	1.46
α-selinene	1499	1498	2.12	α-panasinsene	1764	-	0.34
δ-selinene	1495	1492	0.31	Caryophyllene oxide	1986	1986	-
spathulenol	1585	1577	0.33	(E)-cinnamaldehyde	2031	2033	0.93
Caryophyllene oxide	1589	1582	2.33	(E)-mehtyl cinnamate	2056	2075	13.82
benzyl benzoate	1772	1759	0.17	(E)-ethyl cinnamate	2110	-	18.75
Total amount of identified compounds			98.52				98.35

^a^ Calculated retention index on a DB-5MS capillary column; ^b^ Retention indices on a DB-5MS column from reference (Adams 2012) [35]; ^c^ Calculated retention indices on a HP-INNOWax capillary column; ^d^ Retention indices on a HP-INNOWax column from reference (Babushok 2011) [36].

**Table 2 molecules-28-01741-t002:** Antioxidant activity as IC_50_ DPPH and IC_50_ ABTS of *C. brownei* essential oil, natural control *T. vulgaris* essential oil, and (BHA) chemical control. Mean ± SD (*n* = 3), *p* < 0.05.

Essential Oils/Antioxidant Molecule	DPPH IC_50_ (mg/mL)	ABTS IC_50_ (mg/mL)
*Clinopodium brownei*	1.771 ± 0.260	0.060 ± 0.002
*Thymus vulgaris*	0.010 ± 0.005	0.008 ± 0.002
butylated hydroxyanisole (BHA)	0.004 ± 0.000	0.002 ± 0.000

**Table 3 molecules-28-01741-t003:** MIC values of *C. brownei* essential oil and *T. vulgaris* essential oil (natural control) [22] on various microorganisms. Mean ± SD (*n* = 4), *p* < 0.05.

Microorganism	*C. brownie* MIC (mg/mL)	*T. vulgaris* MIC (mg/mL)
**Gram Positive Bacteria**		
*Staphylococcus aureus* ATCC 6328	7.92 ± 0.21	3.02 ± 0.16
*Enterococcus faecalis* ATCC 29212	5.54 ± 0.25	2.89 ± 0.14
*Listeria grayi* ATCC 1912	3.84 ± 0.16	2.28 ± 0.11
*Staphylococcus epidermidis* ATCC 14990	13.57 ± 0.21	7.98 ± 0.37
**Gram negative Bacteria**		
*Escherichia coli* ATCC 25922	6.22 ± 0.16	2.13 ± 0.09
*Proteus vulgaris* ATCC 6380	4.62 ± 0.18	2.77 ± 0.08
*Klebsiella oxytoca* ATCC 8724	7.19 ± 0.35	1.30 ± 0.05
*Pseudomonas aeruginosa* ATCC 9027	8.38 ± 0.41	3.10 ± 0.12
**Yeast**		
*Candida albicans* ATCC 10231	3.11 ± 0.10	1.45 ± 0.07

## Data Availability

Data are available in the manuscript.

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
