# Peer review of "Chemical Composition, Antimicrobial and Antioxidant Bioautography Activity of Essential Oil from Leaves of Amazon Plant Clinopodium brownei (Sw.)"

_molecules, 2023, doi:10.3390/molecules28041741_

Round 1

Reviewer 1 Report

Dear authors, I have reviewed the document "Bioautography antimicrobial and antioxidant of Clinopodium brownei (Sw.) Essential Oil from Ecuadorian Amazon". Regarding the standard design according to the template of the publisher and journal it changes a little, but of course this design is being accepted more and more. Authors should consider the following before continuing with the processing:

- The title of the document is terse, it should be rewritten, it is not necessary to tell everything in the title, but it is not necessary that it is not very delimited either. 

- Do not repeat the words of the title in the keywords.

- In the summary, before narrating the objective, it is necessary to add a sentence about the introduction, you can write about biodiversity in the study area (Ecuadorian Amazon) or directly about the subject of the study (species).

- Add in the first paragraph about two sentences about biodiversity in Ecuador, you can look for a document in the diversity magazine putting "Biodiversity Ecuador" or "SNAP Ecuador".

- Paragraph two and three should be grouped into one.

- I consider that the last paragraph of the introduction is extensive but not very structured. I recommend that you narrate the problem or why the interest arises to carry out the study and then narrate the objective and how it will contribute to science.

- It would be interesting to read why the difference in comparison with other regions. What is the reason for this?

Author Response

REVIEWER 1

Observation 1. The title of the document is terse, it should be rewritten, it is not necessary to tell everything in the title, but it is not necessary that it is not very delimited either. 

New proposal (title):  Chemical composition, antimicrobial and antioxidant bioautography activity of essential oil from leaves of amazon plant Clinopodium brownei (Sw.)

Observation 2. Do not repeat the words of the title in the keywords.

Keywords: Clinopodium brownei; caryophyllene; cinnamic acid derivates; antioxidant activity; antimicrobial activity

New Keywords: Clinopodium brownei; caryophyllene; cinnamic acid derivates; bioactivity bioautography

Was deleted antioxidant activity and antimicrobial activity and replaced with bioactivity bioautography.

In our opinion the scientific name of plant must remain in the keywords.   

Observation 3. In the summary, before narrating the objective, it is necessary to add a sentence about the introduction, you can write about biodiversity in the study area (Ecuadorian Amazon) or directly about the subject of the study (species).

Was added this sentence before the objective:

The Amazonian region of Ecuador have a great vegetal biodiversity and your people a millennial ancestral knowhow in the therapeutic use and medicinal of this resources.

Observation 4. Add in the first paragraph about two sentences about biodiversity in Ecuador, you can look for a document in the diversity magazine putting "Biodiversity Ecuador" or "SNAP Ecuador".

The sentence add in the summary is:

This peculiar biogeography of Ecuador has made that its biodiversity is considered as one the most important on the word and be part of the list of 17 megadiverse countries, in Ecuador we have approximately 25.000 plant spices, about 10% of all plants in the word with only 0.06 % of the global land surface [3].

Is added the reference number 3:

  1. Mestanza-Ramón, C.; Henkanaththegedara, S. M.; Vásconez Duchicela, P.; Vargas Tierras, Y.; Sánchez Capa, M.; Constante Mejía, D.; Mestanza Ramón, P. In-situ and ex-situ biodiversity conservation in Ecuador: A review of policies, actions and challenges. Diversity, 2020, 12(8), 315.

Observation 5. Paragraph two and three should be grouped into one.

The was executed

The new sentence Paragraph is:

Essential oils are among the secondary metabolites with the greatest potential, and have gained industrial interest due to their simple extraction method [12], easy analysis method of their components [13] and their wide range of biological, pharmaceutical [14-16] and cosmetic [17,18].

Observation 6. I consider that the last paragraph of the introduction is extensive but not very structured. I recommend that you narrate the problem or why the interest arises to carry out the study and then narrate the objective and how it will contribute to science.

The new paragraph is:

All the information about ancestral uses and preliminary studies about C. brownei propose to the plant as a good medicinal resource, for this reason the goal of this research is to value the chemical compounds and the bioactivity in the essential oil with a goal to use pharmaceutical and cosmetics. 

Observation 7. It would be interesting to read why the difference in comparison with other regions. What is the reason for this?

We don’t understand this question. The observation is in the section? Introduction or Results?

Reviewer 2 Report

Please insert HPTLC plate photograph before derivatization with DPPH.

Did you scan the plate after development?

Please improve the experimental section of the GC-MS analysis.

Is the HPTLC solvent system and method reproducible? How you can say without performing method development?

Is the GC-MS method reproducible? How you can say without performing method development?

Has anyone separated ethyl cinnamate and methyl cinnamate through TLC/HPTLC? 

Section 4.8: Concentration of DPPH with reference? 

Author Response

REVIEWER 2

Obesrvation 1. Please insert HPTLC plate photograph before derivatization with DPPH.

The photograph before revelation with DPPH was insert as part of figure 1.

New figure 1 is:

Observation 2. Did you scan the plate after development?

The HPTLC plates, was scanned before and after revealed with a photograph system of our laboratory.  

Observation 3. Please improve the experimental section of the GC-MS analysis.

To improve the understanding of the GC/MS analysis, the technique was separated into sections (paragraphs).

Section a: Columns and equipment

Section b: GC conditions

Section C: mass spectrometer conditions.

The finally 4.4 GC/MS analyses is:

A first analysis was made a column Termo Scientific TR-5MS (5% -phenyl-95% dimethyl) polysiloxane), with a length of 30 m, a thickness of 0.25 mm and a film thickness of 0.25 m. A chromatograph Trace 1310 coupled to mass spectrometer was using.

The operating conditions was: analysis started at 60 °C for 5 minutes, reaching 100 °C at a rate of 2 °C per minute, then it reached 200 °C at a speed of 3 °C per minute, after it reaches 230 °C at a rate 5°C per minute, staying at this temperature for 5 minutes for a total analysis time of 60 minutes.

Mass spectrometer conditions were: ionization energy: 70 eV; emission current: 10 µAmp; scan rate: 1 scan/s; mass range: 40–350 Da; trap temperature: 230 °C; Transfer line temperature: 200 °C [51].

Second analysis was made with an Agilent DBWax column (polyethylene glycol), with a length of 20 m, a thickness of 0.10 mm and a film thickness of 0.20 m. A chromatograph EVOQ 436 GC - TQ coupled to mass spectrometer was using.

The operating conditions was: analysis started at 50 °C for 2 minutes, reaching 80 °C at a rate of 5 °C per minute, then it reached 250 °C at a speed of 5 °C per minute, staying at this temperature for 1 minutes for a total analysis time of 49.50 minutes.

Mass spectrometer conditions were: ionization energy: 70 eV; emission current: 10 µAmp; scan rate: 1 scan/s; mass range: 35–400 Da; trap temperature: 220 °C; Transfer line temperature: 260 °C [52].

Observation 4. Is the HPTLC solvent system and method reproducible? How you can say without performing method development?

The HPTL is a reproduction of a method describe for Noriega et al JPC-Journal of Planar Chromatography-Modern TLC, 31(2), 163-168. The reference was added in this section and section of Antimicrobial bioautography.

Observation 5. Is the GC-MS method reproducible? How you can say without performing method development?.

The method GC-MS is a partial reproduction of this researches:

Saltos et al [51] ( Bol Latinoam Caribe Plant Med Aromat 21 (4): 455 - 463 (2022).  Matailo, A., Bec, N., Calva, J., Ramírez, J., Andrade, J. M., Larroque, C., ... & Armijos, C. (2020). Selective BuChE inhibitory activity, chemical composition, and enantiomer content of the volatile oil from the Ecuadorian plant Clinopodium brownei. Revista Brasileira de Farmacognosia, 29, 749-754

Balcerzak, L., Gibka, J., Sikora, M., Kula, J., & Strub, D. J. (2019) [52]. Minor constituents of essential oils and aromatic extracts. Oximes derived from natural flavor and fragrance raw materials–Sensory evaluation, spectral and gas chromatographic characteristics. Food chemistry, 301, 125283.

The references were added in the manuscript. Differences depends of our columns and equipment.

Observation 6. Has anyone separated ethyl cinnamate and methyl cinnamate through TLC/HPTLC? 

Exist the preliminary evidence of separate ethyl cinammante from Ocotea quixos essential oil (Noriega et al JPC-Journal of Planar Chromatography-Modern TLC, 31(2), 163-168)

For methyl cinnamate from Ocimun Basilicum essential oil (Chemistry and Materials Research Vol.8 No.6, 2016)

This researches show the possibility separation of this molecules from essential oils.

The references were added in the manuscript.

Observation 7.  Section 4.8: Concentration of DPPH with reference? 

Was added the reference: (Noriega et al JPC-Journal of Planar Chromatography-Modern TLC, 31(2), 163-168). In this research was used DPPH solution (1%)

Round 2

Reviewer 1 Report

The authors have complied with the requirements. The document can be published.

Author Response

Observation: Moderate English changes required

All the manuscript was corrected by English department of Salesian Polytechnic University.

Reviewer 2 Report

1. Captions of Figures 1 and 3 must be elaborated.

2. Please mention the percentage inhibition for anti-oxidant activity via DPPH and ABTS assay in the results section.

3. HPTLC directed bioautographic assays does not provide sufficient information on chemical and structural identification of detected compounds rather TLC-MS bioautography needs to be done for accurate identification of detected compounds.

4. Please add the mass spectra of the detected compounds in anti-oxidant and anti-microbial TLC bioautographic assays.  

5. Grammatical errors are still present in the manuscript.

Author Response

Reviewer number 2

Observation 1

Captions of Figures 1 and 3 must be elaborated.

The new captions are:

Figure 1. DPPH Bioautographic assay of Clinopodium brownei essential oil, bioactive molecules: Rf 0.633 methyl cinnamate, Rf 0.709 ethyl cinnamate and Rf 0.937 caryophyllene.

Figure 2. Bioautographic antibacterial assay of Clinopodium brownei essential oil, bioactive molecula Rf 0.937 caryophyllene.

Captions similar to (Molecules 2019, 24(8), 1637)

Observation 2

Please mention the percentage inhibition for anti-oxidant activity via DPPH and ABTS assay in the results section.

For respond this observation was added the paragraph: The IC50 value were calculated from variation of inhibition percentage for each concentration of essential oil in contact of radical’s DPPH and ABTS.

The new section 2.3 Antioxidant activity is:

The results of antioxidant activity by the DPPH and ABTS methods, evaluated as the inhibition percentage at 50%, can be seen in Table 2. The IC50 value were calculated from variation of inhibition percentage for each concentration of essential oil in contact of radical’s DPPH and ABTS. The essential oil of T. vulgaris (natural control) [19] and the chemical control butylated hydroxyanisole (BHA) were used as activity standards, showing the differences in activity with the natural positive standard and the chemical control.

Observation 3

HPTLC directed bioautographic assays does not provide sufficient information on chemical and structural identification of detected compounds rather TLC-MS bioautography needs to be done for accurate identification of detected compounds.

Indeed, the assay is a HPTLC/MS. At the end of section 4.8 the following statement explains what we did.

From these active fractions, the compounds were isolated by dissolving them in dichloromethane. The GC/MS assay was performed under the same conditions, as described in section 4.4 for TR-5MS column.

Observation 4.

Please add the mass spectra of the detected compounds in anti-oxidant and anti-microbial TLC bioautographic assays.

Mass spectra of bioactive molecules detected from C. brownei were:

Methyl cinnamate, ethyl cinnamate and caryophyllene

These spectra will be including as complementary material for the manuscript.

Observation 5

Grammatical errors are still present in the manuscript.

All the manuscript was corrected by English department of Salesian Polytechnic University.
